# Transforming Cross-Linked Cyclic Dimers of KR-12 into Stable and Potent Antimicrobial Drug Leads

**DOI:** 10.3390/biomedicines11020504

**Published:** 2023-02-09

**Authors:** Taj Muhammad, Adam A. Strömstedt, Sunithi Gunasekera, Ulf Göransson

**Affiliations:** Pharmacognosy, Department of Pharmaceutical Biosciences, Biomedical Centre, Uppsala University, Box 591, SE-75124 Uppsala, Sweden

**Keywords:** antimicrobial peptides, backbone cyclization: cathelicidin, LL-37, KR-12

## Abstract

Is it possible to enhance structural stability and biological activity of KR-12, a truncated antimicrobial peptide derived from the human host defense peptide LL-37? Based on the mapping of essential residues in KR-12, we have designed backbone-cyclized dimers, cross-linked via a disulfide bond to improve peptide stability, while at the same time improving on-target activity. Circular dichroism showed that each of the dimers adopts a primarily alpha-helical conformation (55% helical content) when bound to lyso-phosphatidylglycerol micelles, indicating that the helical propensity of the parent peptide is maintained in the new cross-linked cyclic form. Compared to KR-12, one of the cross-linked dimers showed 16-fold more potent antimicrobial activity against human pathogens *Pseudomonas aeruginosa*, *Staphylococcus aureus*, and *Candida albicans* and 8-fold increased activity against *Escherichia coli*. Furthermore, these peptides retained antimicrobial activity at physiologically relevant conditions, including in the presence of salts and in human serum, and with selective Gram-negative antibacterial activity in rich growth media. In addition to giving further insight into the structure–activity relationship of KR-12, the current work demonstrates that by combining peptide stabilization strategies (dimerization, backbone cyclization, and cross-linking via a disulfide bond), KR-12 can be engineered into a potent antimicrobial peptide drug lead with potential utility in a therapeutic context.

## 1. Introduction

A large number of antimicrobial peptides (AMPs) with promising activities have been described to date. However, translation into clinics has been hampered due to challenges associated with the in vivo environment. In humans, LL-37, the only representative member of the cathelicidin-derived AMP, is found in a large variety of cells, tissues, and body fluids [1,2]. LL-37 kills bacteria by binding to the cell membrane, and when a threshold concentration is reached, it co-localizes to form pores, leading to cytoplasmic leakage and cell death [3]. In addition to exerting direct antimicrobial activity, it also binds to and neutralizes bacteriotoxins such as bacterial lipopolysaccharides (LPS) and lipoteichoic acids, [4,5]. LL-37 also modulates immune responses against invading microbes [3,6].

Because of these properties, LL-37 has long been a candidate for development into therapeutics as an immunomodulatory and antibiotic drug. It is currently being pursued in preclinical and clinical settings for topical or local applications [7]. However, some serious obstacles, such as a low synthetic yield, toxicity to human red blood cells [8,9], enzymatic susceptibility [10,11] as well as sub-optimal potency under physiological conditions [12], have hindered the development of LL-37 into a drug lead for systemic use. Despite the above-mentioned challenges, these molecules are attractive alternatives to overcome the alarming problem of antibiotic resistance as adjuvant therapy or to achieve synergistic effects with conventional antibiotics [13,14].

We have previously demonstrated that it is possible to increase the antimicrobial activity of the shortest antimicrobial region of LL-37, known as KR-12, by substituting specific amino acids [15]. In that work, Gln5 and Asp9 were identified as key positions to modulate antibacterial activity. The engineered peptide containing Lys and Ala substitutions at key positions (Q5K and D9A, respectively) displayed up to an 8-fold improvement in antimicrobial activity compared to KR-12. Further engineering of KR-12 and derivatives into head-to-tail cyclic dimers resulted in new peptide analogs with both increased antimicrobial and immunomodulatory activities, coupled with enhanced proteolytic stability [16,17]. When Pro residues were introduced in the linker region, the cyclic dimer CD4-PP exhibited a distinct α-helical profile (78% α-helical content), which was greater than that of parent peptide LL-37 (72% α-helical content) and the cyclic dimer from first-generation cd4 (65% α-helical content) [17].

The objective of the current study was to design cyclic dimers with enhanced structural stability and activity in more physiologically relevant conditions. The design scheme of our current work is depicted in Figure 1. Here, we stabilized the monomeric KR-12 (Q5K,D9A) subunits by introducing a disulfide bond at positions where substitutions are known to enhance the activity [15]. By further incorporating proline residues in the linker regions, we aimed to potentially induce a helix-turn-helix motif in the KR-12 (Q5K,D9A) subunits, to promote an uninterrupted α-helical conformation throughout the subunit structure. Herein, we evaluated the antimicrobial activity of the new cyclic dimers against human pathogens in a two-step microdilution assay. The effects of human serum, various culture media, and salts at physiological concentrations on antimicrobial activity were also examined.

## 2. Results

### 2.1. Design, Synthesis, and Amino Acid Composition of Cross-linked Cyclic Dimer

The peptide template to design the current set of cross-linked cyclic dimers is based on our previous study, in which positions Q5 and D9 were identified as key positions in KR-12 to modulate antimicrobial activity. The dimers were designed by connecting two KR-12 (Q5K,D9A) units via a four-amino acid residue linker. In the cd4-CC peptide, GAGG was used as a linker; and in cd4-CCPP, a proline residue was incorporated within the linker region to promote a turn-helix-turn motif (GPGG). Both the dimers (cd4-CC and cd4-CCPP) were cross-linked via a disulfide bond by replacing Q5 from monomer #1 and D9 from monomer #2 with cysteines. The design of cross-linked cyclic dimers and their amino acid sequence is detailed in Figure 1A–D. All peptides (linear and cyclic) were synthesized based on an established protocol [16]. Oxidative folding of peptides was carried out in an aqueous buffer (0.1 M ammonium bicarbonate, pH 8.2), in which peptides were readily soluble. The net charge, hydrophobicity, and expected and observed masses of the peptides used in this study are given in Appendix A. The analytical HPLC purity and mass spectrometry identity of the synthesized peptides are shown in Appendix A.

### 2.2. Cross-linked Cyclic Dimer Adopts an Alpha-helical Conformation in Membrane Mimetic Environments

To investigate if the helical structure of the monomers is maintained in cross-linked cyclic dimers, they were first analyzed in solution by ^1^H NMR. Limited dispersion and broadened peaks in the NMR spectra demonstrated that dimers were unstructured (90% water and 10% D2O). Therefore, SDS micelles were introduced to the peptide sample in solution to determine whether structural changes occur in an environment mimicking the condition of a bacterial membrane. Since the complexity of the NMR spectra made the assignments impossible, the exact structures under such conditions could not be determined. Additionally, NMR spectroscopy was also performed with lyso-phosphatidylglycerol (PG) micelles. These micelles were intended to mimic the anionic microbial membrane composition, with a micelle diameter equivalent to biological membrane thickness, and were used at a 1:1 peptide-to-micelle ratio. However, the spectral quality did not improve and the overlap of peaks was persistent, resulting in sequence-specific resonance assignment not being successful (Appendix A).

In contrast, analysis via circular dichroism (CD) spectroscopy revealed secondary structure characteristics of the cyclic dimers. All peptides adopted a clear random coil conformation in Tris buffer (Figure 2A,B). In lyso-phosphatidylglycerol micelles, the CD spectra of peptides exhibited a distinct α-helical profile, where the cross-linked dimers had 50–55% α-helical content. The α-helical content (%) of each peptide in Tris buffer and micelles is shown in Figure 2C.

### 2.3. Quantifying the Membrane-disrupting Mechanism of Cross-linked Cyclic Dimers

A liposome assay was used to identify and quantify mechanism of action of the cross-linked cyclic dimers. An *E. coli* polar lipid extract and a POPC/cholesterol system were used in the form of large unilamellar liposomes to mimic bacterial or human cell membranes, respectively. As demonstrated in Figure 3A, the relative level of membrane leakage within each lipid system mirrored the order exhibited in the two-step microdilution assay (i.e., Figure 3) with regard to the antimicrobial activities of these peptides. The monomer peptides induced leakage from both the lipid systems but with much-reduced potency as compared to LL-37 and cross-linked cyclic dimers. The cyclic dimers induced substantial leakage from *E. coli* with an EC50 threshold lower than LL-37 (Figure 3B POPC/cholesterol system). This supports that the mechanism of action is membrane permeabilization and that difference in activity between peptide variants is directly proportional to the ability of each peptide to bind and disrupt the corresponding membrane. On POPC/cholesterol system, the human model membrane system, LL-37 seems more potent as compared to the cyclic dimers (Figure 3B).

### 2.4. Cross-Linked Cyclic Dimers Are More Potent Than LL-37 in a Two-step Microdilution Assay

All the peptides were subjected to a two-step microdilution assay to determine the minimum inhibitory concentration (MIC) against four strains of human pathogens: the Gram-negative *Escherichia coli* and *Pseudomonas aeruginosa,* the Gram-positive *Staphylococcus aureus,* and the fungus *Candida albicans.* Figure 4 shows the relative MICs of the cross-linked dimers compared to the MIC of KR-12. Both the dimers showed potent activity as compared to KR-12, KR-12 (Q5K,D9A), and LL-37 peptides against *P. aeruginosa*, *S. aureus,* and *C. albicans.* Notably, the activity of the cross-linked cyclic dimers surpassed the activity of LL-37. A complete list of median MIC values is available in Table 1.

### 2.5. Influence of Physiological Salt Concentrations on the Antimicrobial Activity of Cross-linked Cyclic Dimers

The antimicrobial activity of the cross-linked cyclic dimers was tested in the presence of monovalent, divalent, and trivalent salts (150 mM NaCl, 6 μM NH_4_Cl, 4.5 μM KCl, 2.5 mM CaCl_2_, 1 mM MgCl_2_, 4 μM FeCl_3_, i.e., physiological ionic strength) as shown in Table 1. The linear peptides LL-37, KR-12, and KR-12 (Q5K,D9A) were inactive at the highest tested concentrations against *C. albicans* in the presence of NaCl, CaCl_2,_ and MgCl_2_. In contrast, the cross-linked cyclic dimers were active, but their activity was 4-to 16-fold attenuated, as compared to the salt-free MIC from the two-step microdilution assay. Notably, cross-linked cyclic dimers retained antifungal activity with only a 2-fold decrease in activity against other salts (KCl, NH_4_Cl, and FeCl_3_). The inhibitory effect of salts against *S. aureus* was mainly due to NaCl and CaCl_2_, i.e., the cyclic dimers-maintained activity in the presence of other salts. The linear peptides, KR-12 and KR-12 (Q5K,D9A) became inactive at the highest tested concentrations (i.e., 10 µM) in the presence of NaCl, CaCl_2,_ and KCl against *S. aureus.* We found that the presence of salts has little effect on the antibacterial activities of the cross-linked cyclic dimers as well as LL-37 peptide against Gram-negative strains (*E*. *coli* and *P. aeruginosa*). In contrast, KR-12 and KR-12 (Q5K,D9A), both lost activity within the tested concentration range in the presence of NaCl, CaCl_2,_ and MgCl_2_.

### 2.6. Cross-linked Cyclic Dimers Retained Antimicrobial Activity in the Presence of Human Serum and Rich Growth Media

Inactivation of antimicrobial peptides in human serum as well as standard growth media has hindered efforts to develop many AMP-based drug leads for systemic use. Linear peptides including LL-37 were not active in 25 % human serum at 40 μM as shown in Table 1. In contrast, the cross-linked cyclic dimers retained much of their antimicrobial activity with a 4- to 8-fold increase in MIC against Gram-negative strains and a 16- to 32-fold increase in MIC against Gram-positive strain, *S. aureus,* if we compare with the salt-free MIC from the two-step microdilution assay (Table 1).

Notably, linear peptides, KR-12, and KR-12 (Q5K,D9A) were inactive at the highest tested concentration of 80 μM against both Gram-positive and Gram-negative bacteria in the microdilution assay (using 2.1% MHB) (Table 1). LL-37 and cyclic dimers retained activity against Gram-negative bacteria with a 4- to 8-fold increase in MIC (*E. coli* 2.5 μM and *P. aeruginosa* 5 μM). None of the peptides were active against the Gram-positive *S. aureus* strain at the highest tested concentrations for linear peptides (80 μM) and cross-linked cyclic dimers (40 μM).

To determine whether the observed loss in antimicrobial activity against Gram-positive strain *S*. *aureus* occurs due to the MHB medium in general or if it was species-specific effect, we evaluated the antimicrobial activity against two other Gram-positive strains, *B. subtilis* and *B. cereus,* in the presence of standard MHB and also TSB media (MIC reported in Appendix A). Notably, all peptides including cyclic dimers lost activity also against *B. subtilis* and *B. cereus* in both rich growth media. The consistent behavior of the cyclic peptides against Gram-positive bacteria inclined us to presume the reasons as morphological adaptations in the bacterial membrane when grown in rich growth media. However, such changes are unlikely to occur instantaneously, which is why a transfer to a minimum media condition immediately prior to administration of these fast-acting antimicrobial agents is relevant in order to isolate the cause of inhibition and perhaps achieve a more relevant activity measurement [18].

### 2.7. Bacteria Are More Susceptible to Cross-linked Cyclic Dimers in Ionic Environment

It has been reported previously that several AMPs, such as cathelicidins and members of the defensin family lose antimicrobial activity in vitro with the addition of salts at physiological concentration, plasma proteins, or under culture conditions applied to conventional antibiotics. [12,19]. Additionally, the balance of host ionic conditions is known to dictate microbial sensitivity to LL-37 [20]. To understand the contributing factors for the curtailed activity in rich media, we evaluated the antimicrobial activity under conditions that mimic the in vivo (ionic environment), by incorporating bicarbonate ions (50 mM NaHCO_3_) and evaluated the antimicrobial activity. Both dimers retained the antibacterial activity against Gram-negative and Gram-positive strains (i.e., cd4-CCPP, MIC = 0.625 μM against *S. aureus*) (Table 1). Of the panel of tested peptides, only KR-12 failed to show any activity even at the maximum tested concentration of 20 μM.

### 2.8. Cross-linked Cyclic Dimers Are Active against LL-37-resistant Strains

To evaluate the efficacy of cross-linked cyclic dimers against AMP-resistant bacteria, we tested their antimicrobial activity against *Salmonella enterica* serovar *Typhimurium* (wild-type DA6192), *Salmonella enterica* LL-37-resistant mutants (DN-22427-waaY and DN-23307-phoP), *E. coli* MG1655 (wild type) and *E. coli* cyclotides-resistant mutants (DA54114 and DA57105). Both cyclic dimers exhibited higher antimicrobial activity (4- to 8-fold reduction in MIC) against all the tested *S. typhimurium* strains than LL-37 (Table 2). The dimers showed higher activity as compared to KR-12, LL-37, and KR-12(Q5K, D9A), against *E. coli* wild-type and cyclotides-resistant mutants. The cyclic peptide cd4-CCPP was 4-fold more active as compared to LL-37 against *resistant* mutants.

### 2.9. Cross-linked Cyclic Dimers Are Less Hemolytic

The hemolytic activity of the peptides against human red blood cells (RBCs) was determined as a measure of their toxicity to mammalian cells. At the highest test concentrations (80 μM), the cyclic dimers and LL-37 induced 19–23% hemolysis, while the monomers (KR-12 and KR-12 (Q5K,D9A)) only induced 1–4% hemolysis.

### 2.10. Protease Resistance of Cross-linked Cyclic Dimers

One of the major limitations of AMPs for clinical use is their inactivation by both endogenous human proteases and proteases secreted from invading microbes. From our previous studies, it was evident that KR-12 and truncated analogs degrade within 10 min in diluted human serum [15]. More recently, we have shown that the KR-12 cyclic dimer peptides were stable for up to 6 h in diluted human serum but the recovery of LL-37 was very low during the precipitation step of the assay, either due to binding of LL-37 with serum components or rapid degradation [16]. We also compared the stability profile of cross-linked cyclic dimers in diluted human serum. The recovery of the tested peptides (cd4-CC and cd4-CCPP) was very low as compared to peptides in PBS. The low recovery of the cyclic peptides was most probably due to the precipitation of the peptides with serum proteins.

The peptide stability was also assessed in the presence of aureolysin from *S. aureus* at different time points. LL-37 peptide was susceptible to enzymatic degradation immediately after incubation, it was evident by hydrolysis of the Leu31-Val32 peptide bond within the first 5 min. The cyclic peptide, cd4-CCPP was stable for up to 8 h (Figure 5). The negative control, cyclotide Kalata B7, was stable in aureolysin for up to 24 h. Taken together, these experiments established that cross-linked dimers are more stable than linear counterparts to protease degradation by either a single pure protease or a mixture of proteases in human serum.

## 3. Discussion

In the current work, cross-linked cyclic dimers were designed and their antimicrobial activity was evaluated under physiologically relevant conditions proven to be challenging for many of the most commonly studied AMPs. Dimers maintained antibacterial activity in physiologically relevant conditions and rich growth media. Notably, the dimers showed exceptional proteolytic stability as compared to parent peptides. Moreover, cyclic peptides demonstrated activity against various AMP-resistant bacterial strains and retained alpha-helical content in membrane mimicking environment.

In the last 30 years, continuous efforts have been made to develop AMPs as broad-spectrum antimicrobials, either alone or in combination with conventional antibiotics, or as immunomodulatory or LPS-neutralizing compounds [21,22]. For a peptide antibiotic to be considered for therapeutic development, it needs to possess potent, preferably selective antimicrobial activity, low toxicity to human cells, as well as low susceptibility to degradation by proteases of both human and bacterial origin. Thus, chemical strategies that can be incorporated into peptide chains, which can induce helicity and structural stability to improve cell selectivity and increase resistance to proteases are of great interest.

On this ground, we reasoned that introducing disulfide connectivity within a head-to-tail cyclized backbone would bestow improved structural stability to KR-12 peptide-based dimers. Two KR-12 (Q5K,D9A) monomers were joined together into a consistent cycle of peptide bonds, utilizing two four-residue amino acid linkers. In the cd4-CCPP dimer, a Pro residue was incorporated in the four-residue linker region to induce a helix-turn-helix and potentially maximize the native helical structure important for antimicrobial activity. By this design, we hypothesized that backbone cyclization and disulfide bond introduction would increase structural and proteolytic stability, and having two preformed α-helices in close proximity in their active conformation would increase activity.

First, NMR spectroscopy was utilized to determine if the structural design was successful. In both the dimers, due to overlapping signals and poorly separated peaks, it was difficult to assign chemical shifts for the amino acid residues, which we interpreted as an indication of disordered regions. The cross-linked cyclic dimers did not display defined structures either in water or in the membrane-mimicking environment (SDS and Lyso-PG). The introduction of a turn-inducing proline residue in the linker regions, and a disulfide bond between the two monomer subunits, did not improve signal dispersion. Nevertheless, analysis using CD revealed secondary structure: both the dimers were in random coil conformation in buffer, which transformed into broadly α-helical conformation in a membrane-mimicking environment. This result was consistent with previous studies, indicating that a bacterial membrane-like environment is important for LL-37 and KR-12 to form α-helical structure and exert antimicrobial activity [16,23].

The liposome leakage indicated a membrane-disrupting activity for cross-linked cyclic dimers, similar to its parent peptide LL-37 and cyclic dimers peptides from our previous studies. Notably, cd4-CCPP outperformed LL-37 in terms of bacterial target selectivity by a factor of 6-fold. The cd4 dimer from the first-generation cyclic dimers [16], which was designed based on the native KR-12 sequence showed less membrane activity on *E. coli* liposomes as compared to cross-linked dimers (Figure 4A) More recently, we also have reported that LL-37 and CD4-PP caused the formation of blebs on the bacterial surface [17]. It has been reported that the interaction of Gram-negative bacteria with cationic AMPs not only involves components of the bacterial cell wall and lipopolysaccharide but also cell surface receptors [24,25,26]. Taken together, our data suggest that the formation of α-helical structures upon interaction with the bacterial membrane and subsequent membrane permeabilization are key steps for the mechanism of action of cross-linked dimers. However, we cannot rule out other possible targets within the cytoplasm, as previously reported for LL-37 [27].

The relative EC50s between the two lipid systems, for each peptide, may appear inconsistent with the selectivity of cationic AMPs for anionic lipid systems over zwitterionic ones and indeed for self-toxicity of an endogenous host defense peptide (LL-37). Instead, membrane-active LL-37 appear more potent at permeabilizing POPC: cholesterol liposomes rather than those of *E. coli* phospholipids, a phenomenon we believe is linked to the artificial homogeneity of the alkyl chains of the POPC lipids, limiting relevant comparisons to within each lipid system rather than between them in this case. The cross-linked cyclic dimers showed potent antimicrobial activity with an increase of 16-fold as compared to KR-12 against *P. aeruginosa, S. aureus, and C. albicans*, surpassing the activity of the parent peptide LL-37 in a two-step microdilution assay. The relative increase in the activity demonstrates that the dimerization can be seen not only as a 2-fold increase in the number of active peptides in the system but also as an artificial increase in local peptide concentration on the bacterial membrane.

Among the pitfalls in the development of AMPs into therapeutics is their inactivation or loss of antimicrobial activity by physiological salt, human serum, and pH [28]. It has been reported previously that various salt ions affect the antimicrobial activity of AMPs by interfering with electrostatic interactions and triggering competition in membrane binding between cations and peptides. Different chemical strategies have been used to design salt-resistant short helical AMPs, e.g., by introducing helix-capping motifs and also by the insertion of β-amino acids [29,30,31]. The present study investigated the sensitivity of cross-linked cyclic dimers to salts at physiological concentrations by monitoring the changes in MIC values. The antibacterial activity of the cross-linked dimers was only marginally affected by the presence of Na^+^, NH^4+^, K^+^, Ca^2+^, Mg^2+^, and Fe^3+^ ions (Table 1). Although some cations (including Na^+^, Ca^2+^_,_ and Mg^2+^) slightly compromised the antifungal activity of cross-linked cyclic dimers. One of the hypotheses for the salt sensitivity of KR-12, KR-12 (Q5K,D9A) including LL-37 is due to the more flexible skeletal structure which is prone to structural deformity in the presence of various salt ions and subsequent loss of membrane binding and permeabilizing activity. The cross-linked dimers with rigid structural constraints maintained antimicrobial activity, as reported in previous studies [32].

The stability profile of cross-linked cyclic dimers in the presence of human and bacterial proteases was evaluated by testing in human serum and the pure bacterial protease, aureloysin. In previous studies, linear peptides including LL-37 and KR-12 were reported to be unstable in human serum [11,15]. The recovery rate of cross-linked cyclic peptides in the current study was very low compared to the peptides in PBS. To address whether the low peptide recovery could be due to enzyme susceptibility, antimicrobial activity was tested in the presence of human serum. The linear peptides, including the parent peptide, LL-37, completely lost the antimicrobial activity against all bacterial strains when tested in the presence of serum. It has been reported that LL-37 loses antimicrobial activity in the presence of human serum by binding to serum components, i.e., albumin or apolipoprotein A-1 [19]. Likewise, we speculate that the loss-in activity of cyclic peptides was due to binding to serum components as we were not able to identify any cleavage products. Moreover, LC–MS analysis of the linear peptides including the LL-37 reaction mixture with bacterial protease indicated that the peptides were efficiently cleaved. In contrast, the cross-linked cyclic dimers were intact for up to 8 h. It indicates that the relative improvement of stability of cyclic peptides, in the presence of aureolysin, is likely due to backbone cyclization with a cross-linking via a disulfide bond.

Cyclic dimer peptides were furthermore subjected to a series of assays to probe the performance of these peptides for some of the most important challenges in current antimicrobial drug design: non-specific toxicity, reduced activity in environments used for conventional antibiotics, and the emergence of bacterial resistance to AMPs. Unspecific cell toxicity is among the major hurdles in the development of AMPs, particularly when the target of action is the cytoplasmic membrane. The cross-linked cyclic dimers showed low hemolytic activity. This is an improvement compared to previous analogs [16]. Even though it is known that the composition of rich growth media strongly impacts the activity of antimicrobial peptides, including LL-37 [33,34]. Although our primary goal in this study was to examine the properties of cross-linked cyclic peptides in physiologically relevant conditions, we also performed NCCLS-type broth microdilution assays to substantiate the results of our two-step microdilution assay. As predicted, we found that the use of the conventional medium in the microdilution assay underestimates the activity of cyclic peptides. More surprisingly, all the peptides became completely inactive at the highest tested concentration against the Gram-positive strains irrespective of the cultural media used or bacterial species tested. To our surprise, when the activity was tested in the presence of bicarbonate, an ubiquitous molecule in many microenvironments of the human body, both the Gram-positive and Gram-negative bacteria show dramatically increased sensitivity to dimers. It has been suggested previously that bacterial susceptibility to AMPs is significantly higher in the mammalian ionic environment due to significant changes in gene expression [20], which leads to membrane thinning.

AMPs presumably develop resistance at a slow pace in vitro but resistance against one AMP could confer cross-resistance to others or in some cases increase the collateral sensitivity [35,36]. Hence, we tested antimicrobial activity against LL-37 and cyclotide resistant-bacteria. It is noteworthy that cross-linked dimers exhibited more potent antimicrobial activity against all strains than LL-37. The *S. typhimurium* strains used in this study were LL-37-resistant and had mutations in the genes *waaY* and *phoP,* which are connected with lipopolysaccharides (LPS) modifications [35]. The *E. coli* MG1655 strains were resistant to cyclotides (cyO19 and kB7). The *E. coli* DA54114 which was cyO19 resistant had a mutation in gene *prc*, which has a role in osmotic stress. The kB7-DA57107 mutant was un-characterized [37]. The cyclic peptides were 4- to 16-fold more active as compared to linear peptides against cyclotide-resistant strains.

## 4. Materials and Methods

### 4.1. Peptide Synthesis

All the linear and cross-linked cyclic dimers were synthesized as previously described [15,16]. In brief, all the peptides were synthesized following standard Fmoc/tBu chemistry on CEM Liberty 1 automated microwave-assisted peptide synthesizer. For cyclic peptides, backbone cyclization was carried out using the Native chemical ligation (NCL) protocol, and later on, the peptides were oxidized in 0.1 M ammonium bicarbonate, pH 8.2.

### 4.2. Nuclear Magnetic Resonance (NMR) Experiments

Freeze-dried dimer peptides (~2 mg) were dissolved in 220 mL of H2O/D2O (9:1, *v*/*v*) at pH ~5 to acquire one- and two-dimensional NMR spectra (^1^H-^1^H TOCSY, ^1^H-^1^H NOESY) and processed as previously described [16]. Spectra were acquired with and without the addition of either deuterated sodium dodecyl sulfate (SDS; Merck; peptide: SDS 1:40 molar ratio) or lyso-phosphatidylglycerol (lyso-PG) micelles 1 mM, 16:0 lyso-PG (1-palmitoyl-2-dydroxy-sn-glycero-3-phospho-(1′-rac-glycerol) sodium salt by Avanti Polar Lipids. NMR spectra were acquired on a Bruker Avance Noes 600 MHz TCI (CRPHe TR-^1^H and ^19^F/^13^C/^15^N 5mm-EZ) spectrometer.

### 4.3. Circular Dichroism Spectrum Analysis

The α-helical contribution to the secondary structure of the peptides was determined using a JASCO J810 spectropolarimeter (JASCO Corporation, Easton, MD, USA) monitoring changes in the 200–260 nm range at 37 °C in 10 mM Tris-HCl buffer (pH 7.4), with stirring in a 1 cm quartz cuvette. Signals from the peptides, at a concentration of 10 µM, were measured in buffer alone and with 1 mM lyso-PG (1-palmitoyl-2-dydroxy-sn-glycero-3-phospho-(1′-rac-glycerol) sodium salt by Avanti Polar Lipids, Alabama. Each spectrum depicted in Figure 2, is the mean from 10 accumulations at a rate of 50 nm/min. The solvent background contribution was subtracted for each wavelength as well as the baseline drift for each measurement (normalized at 260 nm, where no peptide signal is present). The quantification of α-helix composition was calculated at 225 nm and compared to a poly-L-Lys reference (30–70 kDa) from Sigma-Aldrich, St Louis, MO, USA) in 0.1 M NaOH (100% helix) and 0.1 M HCl (100% coil).

### 4.4. E. coli Liposome Leakage Assay

The liposome leakage assay was performed as described previously [38]. In brief, the following lipids/lipid mixture was used: *Escherichia coli* polar lipid extract or 1-palmitoyl-2-oleoyl-sn-glycerol-3-phosphocoline1-palmitoyl-2-oleoyl-sn-glycerol (POPC, both from Avanti Polar Lipids, Alabaster, US-AL) and cholesterol (Sigma Aldrich). The lipid bilayers were formed on round-bottom flask walls by dissolving lipids in chloroform was evaporated under a flow of N_2_ and the lipid bilayer was placed in a vacuum overnight. Lipid bilayers were re-suspended in an aqueous solution of 100 mM 5(6)-carboxyfluorescein in 10 mM Tris buffer. Due to a shorter interlamellar distance together with a higher resistance to undulations; the zwitterionic, cholesterol-containing composition, POPC/cho (60/40 mol%) requires repeated freeze–thawing. Multilamellar structures and polydispersity were reduced by repeated extrusion through 100-nm polycarbonate membranes and un-trapped carboxyfluorescein was removed by gel separation. Un-trapped carboxyfluorescein was removed by gel filtration on Sephadex PD-10 columns. Membrane permeability was measured by monitoring carboxyfluorescein efflux from the liposomes to the external low-concentration environment, resulting in loss of self-quenching and an increased fluorescence signal with excitation and emission wavelengths of 492 and 517 nm, respectively. The leakage experiments were performed on a 96-well plate format. Wells were prepared with a 2-fold serial dilution of the peptides in Tris buffer. The plates were pre-heated to incubation temperature (37 °C) and liposomes were subsequently administered (to a final lipid concentration of 10 μM in 200 µL) by the automated dispenser. The effect on liposome permeability for each peptide concentration was monitored for 45 min, at which point leakage had largely subsided. The results shown represent the mean from triplicate experiments with standard deviations and are presented as the percent of total leakage generated with Triton X-100 and subtraction of the baseline value. The EC50 values are (when applicable) calculated using sigmoidal dose–response curves, with the leakage percentage (0–100 constraints) as a function of the peptide concentration (log10), using GraphPad Prism.

### 4.5. Bacterial and Fungal Strains were Used

The three bacterial strains—*Escherichia coli* ATCC 25922, *Pseudomonas aeruginosa* ATCC 27853, and *Staphylococcus aureus* ATCC 29213—as well as the fungal strain *Candida albicans* ATCC 90028 were provided by the Department of Clinical Bacteriology, Lund University Hospital. *Bacillus subtilis* CCUG 163B^T^ and *Bacillus cereus* CCUG 7414^T^ were obtained from the Department of Molecular Sciences, Swedish University of Agricultural Sciences.

The LL-37 resistant mutants (DA22427 and DA23307) used in this study were derived from *Salmonella enterica* serovar Typhimurium DA6192, wild-type and cyclotides-resistant mutants (DA54114 and DA57107) were derived from *E. coli* MG1655 (wild type) and were kindly provided by Department of Medical Biochemistry and Microbiology, Uppsala University.

### 4.6. Antimicrobial Peptide Susceptibility Testing

Antimicrobial activities of the peptides were evaluated using the two-step microdilution assay, which is designed for testing AMPs without activity-inhibiting components [18]. Briefly, bacteria were grown to the mid-logarithmic phase at 37 °C in 3% (*w*/*v*) tryptic soy broth (TSB) (Merck KGaA, Darmstadt, Germany). The microbial culture was washed twice by centrifugation and resuspension in 10 mM Tris buffer (pH 7.4 at 37 °C, adjusted by HCl). The microbial suspension was quantified by OD_600_, diluted, and transferred to 96-well untreated polystyrene microplates prepared with 2-fold serial dilutions of the peptides using 50,000 bacteria per well (all dilutions in Tris buffer). These were incubated for 5 h at 37 °C, after which 5 µL of 20% (*w*/*v*) TSB was added to each well and the microplates were re-incubated for an additional 6–12 h (depending on the growth rate of each organism). All MICs presented are the medians from triplicate experiments performed independently. The MIC was defined as the lowest peptide concentration that fully inhibited bacterial growth.

### 4.7. Resistance of Peptides to Salts and Human Serum

A two-step assay was supplemented with a physiological concentration of salts and diluted human serum and the MIC of the peptides was evaluated as in 4.6.

### 4.8. MIC in Mueller–Hinton Broth (MHB) and Ionic Environment

The standard broth microdilution assay was utilized to assess the MIC of designed peptides according to the guidelines of the Clinical and Laboratory Standards Institute (CLSI) [18]. Briefly, bacteria were grown in unbuffered Mueller–Hinton Broth (MHB, VWR International, Leuven, Belgium) as well as buffered (containing 50 mM NaHCO_3_) medium to the mid-logarithmic phase and were quantified by OD_600_. The bacterial suspension was diluted in the same medium to reach a final density of 1 × 10^6^ colony-forming unit per milliliter (CFU/mL). A volume of 50 µL of the bacterial suspension was added to 96-well plates containing 50 µL of MHB in the absence (viability control) or the presence of the peptides at different concentrations (0.625–80 µM). MIC was defined as the lowest concentration of each peptide resulting in the complete inhibition of visible growth after 24 h of incubation at 37 °C. MIC values were derived from 3 sets of independent experiments with different bacterial cultures.

### 4.9. MIC against Resistant Strains

MIC measurements were performed in 20 mM sodium phosphate buffer (PBS), supplemented with 0.1% TSB in 96-well plates (round bottom, Nunc A/S, Roskilde, Denmark) in a total volume of 100 µL. The overnight cultures were diluted 5 × 10^5^ CFU/mL in the same medium and 90 µL was transferred into the plates containing 10 µL of 2-fold diluted peptides. The plates were incubated for 16 to 20 h at 37 °C and the MIC was determined by visually examining the plate. At least two independent experiments were performed with two technical replicates. In cyclotides, resistant mutant average MIC was reported.

## 5. Resistance to Proteolytic Digestion

### 5.1. Stability Human Serum

Peptide stability was assayed in diluted human serum as described in an earlier study [15]. Briefly, 25% (vol/vol) human serum was centrifuged at 13,000 rpm for 10 min to remove lipids and the supernatant was collected and incubated at 37 °C for 15 min. Here, 8 µL of selected aqueous peptide stock solutions (200 µM) were incubated in diluted human serum (80 µL) at 37 °C for the desired time points (0, 2, 4, 6, and 24 h for cross-linked cyclic peptides). 80 µL of 20% trifluoro acetic acid (TCA) was added and incubated at 4 °C for 10 min. Then, 80 µL of 6 M urea was added and incubated at 4 °C for a further 10 min. The samples were centrifuged at 13,000 g for 10 min and the supernatant 30 µL was analyzed on MS coupled to RP-HPLC.

### 5.2. Stability in Commercial Proteases

Peptide stability was also assayed in commercially available Gram-positive, *S. aureus* protease, aureolysin. A volume of 8 µL of (200 µM) peptide solution was mixed with 2 µL (5 µM) of enzyme solution, hence the final peptide: enzyme ratio was 200:1 in 40 µL. All the samples were then incubated at 37 °C at different incubation times. The reaction was quenched by acidifying with 5 µL of 0.1% formic acid. The samples were analyzed on MS coupled to RP-HPLC.

### 5.3. Hemolysis Assay

The hemolytic assay was conducted by preparing a stock solution of 320 μM peptides in MilliQ water and serially diluted in PBS to give 100 μL test solutions in a 96-well U-bottom microtiter plate (Nunc). Human-type RBCs (red blood cells) were collected from the donor, washed with PBS (pH 7.4) three times, and centrifuged at 1500× *g* for 60 s. A pellet of RBCs was resuspended to 0.25% (*v*/*v*) solution in PBS (pH 7.4). A 100 μL 0.25% suspension of washed RBCs in PBS was dispensed into each well containing serially diluted peptide solutions and incubated at 37 °C for 1 h. The 96-well plate was centrifuged at 150× *g* for 5 min and aliquots of 100 μL were transferred to a 96-well U-bottomed microtiter plate (Falcon) and the absorbance was measured at 405 nm with an automatic Multiskan Ascent plate reader (Labsystems). The amount of hemolysis was calculated as the percentage of maximum lysis (1% Triton X-100 control) after adjusting for minimum lysis (PBS control).

## 6. Conclusions

We have demonstrated that cross-linked cyclic dimers were able to overcome the challenges associated with human serum and physiological concentrations of different salts, while linear counterparts displayed a high sensitivity. The findings from this study suggest that multiple parameters need to be assessed in the design and development of AMP-based antimicrobials, for local as well as systemic applications. It is also recommended that analysis of microbial susceptibility to AMPs should be performed with organisms grown in an environment corresponding to the mammalian host for which they represent a potential pathogen.

## Figures and Tables

**Figure 1 biomedicines-11-00504-f001:**
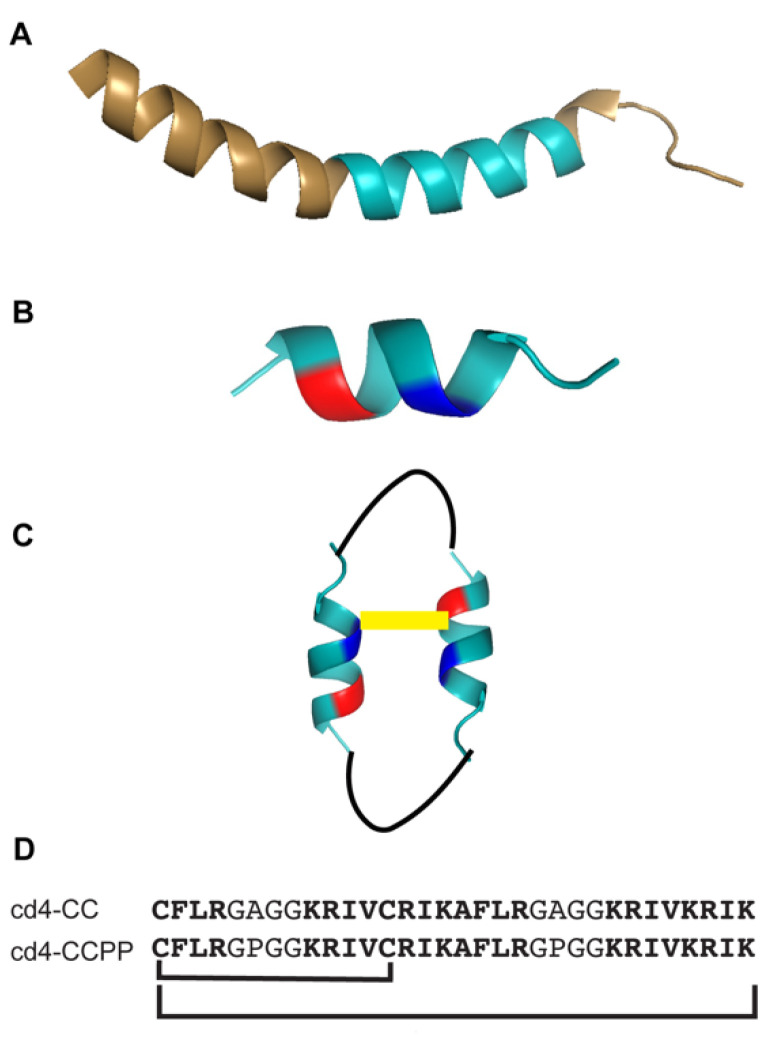
The design strategy of cross-linked cyclic dimers. The dimers are based on the KR-12 monomer, which is an internal segment of LL-37. (**A**) The three-dimensional structure of LL-37 (PDB code 2K60). (**B**) The optimized peptide KR-12 (Q5K,D9A) from previous studies. (**C**) A schematic representation of the cross-linked cyclic cd4-CCPP. (**D**) Peptide sequences and the head-to-tail cyclization (between lysine and cysteine) and the disulfide bond highlighted by bold connecting lines.

**Figure 2 biomedicines-11-00504-f002:**
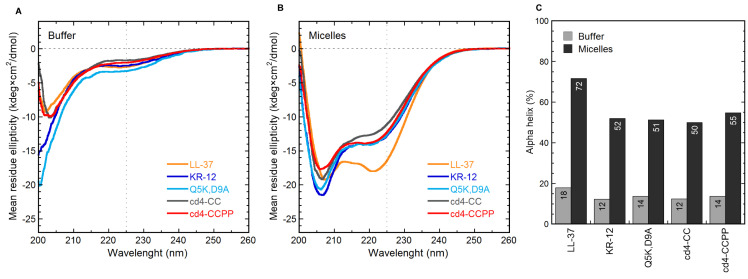
Peptide structure from circular dichroism spectrometry. The mean residue molar ellipticity curves from LL-37 and cross-linked cyclic dimers. (**A**) peptides in Tris buffer only and (**B**) peptides in 16:0 lyso-phosphatidylglycerol (lyso-PG) micelles at a 1:1 peptide-to-micelle ratio (**C**) the amount of alpha helix, The mean residue ellipticity was plotted against wavelength. The amount of α-helix is determined from the signal intensity at 225 nm using a poly-Lys reference.

**Figure 3 biomedicines-11-00504-f003:**
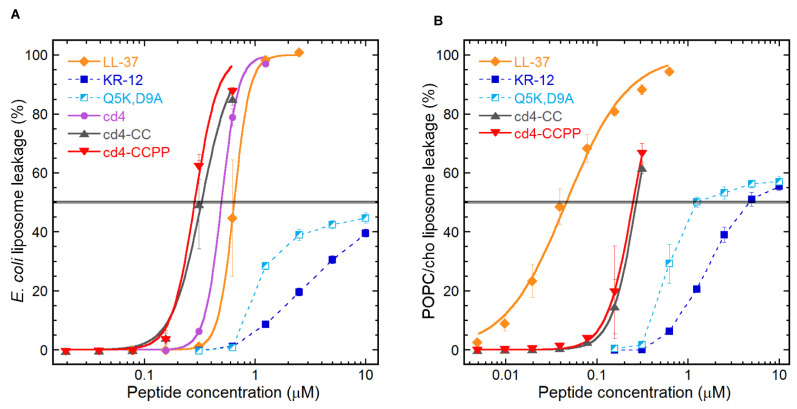
Liposome leakage levels as a function of peptide concentration. The fluorescent marker (carboxyfluorescein) efflux levels from liposomes after 45 min of incubation with peptides is presented. The liposome systems used were: (**A**) *E. coli* polar lipid extract as a generic bacterial composition and (**B**) POPC: cholesterol (3:2 molar ratio) as a simplified human model. Results are the means from triplicate experiments with standard deviations.

**Figure 4 biomedicines-11-00504-f004:**
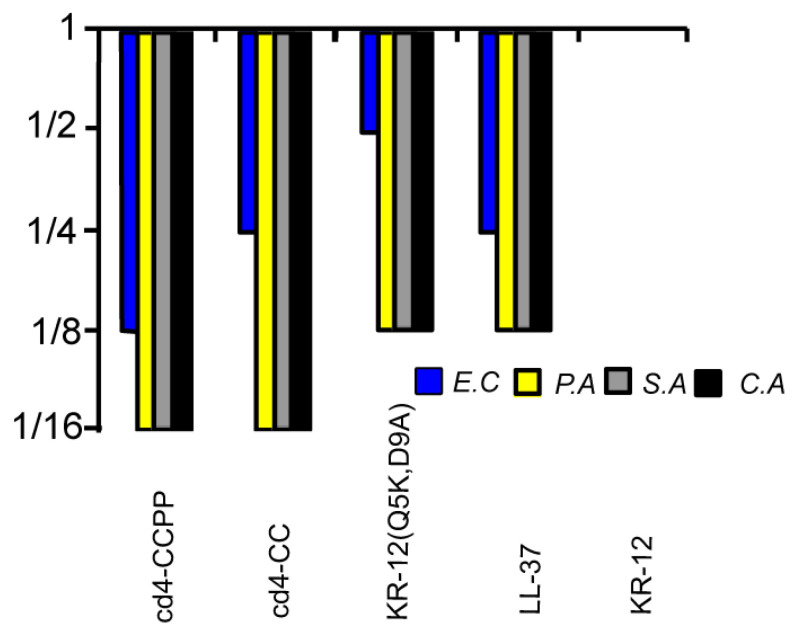
Relative MICs of the cross-linked cyclic dimers. The colors/patterns represent the organisms: *E. coli* (blue), *P. aeruginosa* (yellow) *S. aureus* (gray), and *C. albicans* (black). A four-residue linker was used to connect two KR-12 (Q5K,D9A) units. MIC values of LL-37 are *E. coli*: 0.625 µM; *P. aeruginosa*: 1.25 µM; *S. aureus*: 1.25 µM; *C. albicans* 2.5 µM. For cd4-CCPP values are 0.312, 0.625, 0.625, and 0.625 µM, respectively. All median MIC values can be found in Table 1.

**Figure 5 biomedicines-11-00504-f005:**
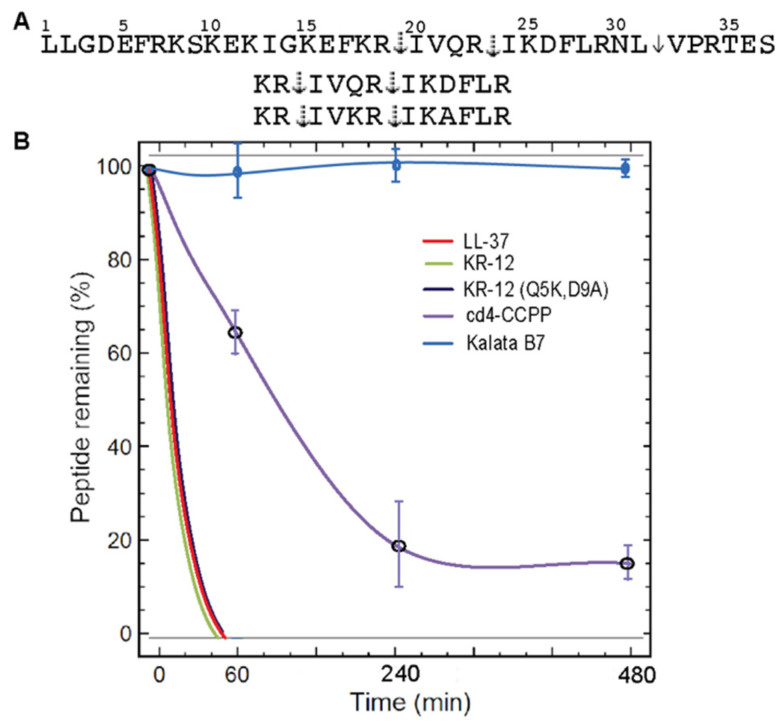
Proteolytic stability of cyclic peptides in relevant biological conditions and cleavage sites of parent peptides (LL-37, KR-12, KR-12 (Q5K,D9A)) by aureolysin. (**A**) Solid arrows indicate cleavage sites identified in this study, and dotted arrows represent cleavage sites reported earlier by aureolysin. (**B**) Stability profile of peptides in the presence of *S. aureus* protease, aureolysin.

**Table 1 biomedicines-11-00504-t001:** MIC values in μM of the peptides in the presence of physiological salts, human serum, rich growth media (MHB), and ionic environment.

Strain/Peptides	Control ^1^	NaCl	NH_4_Cl	KCl	CaCl_2_	MgCl_2_	FeCl_3_	25% Serum	MHB ^2^	IE ^3^
** *E. coli* **										
LL-37	0.625	5	1.25	1.25	>10	5	2.5	>40	20	10
KR2-12	2.5	>10	10	10	>10	>10	10	>40	80	>20
KR-12 (Q5K,D9A)	1.25	>10	5	2.5	>10	>10	2.5	>40	80	10
cd4-CC	0.625	2.5	2.5	1.25	2.5	1.25	2.5	5	5	1.25
cd4-CCPP	0.312	2.5	1.25	0.625	1.25	1.25	1.25	2.5	5	0.625
** *P. aeruginosa* **										
LL-37	1.25	5	1.25	1.25	>10	5	2.5	>40	20	10
KR-12	10	>10	>10	10	>10	>10	10	>40	80	>20
KR-12 (Q5K,D9A)	1.25	>10	5	2.5	>10	>10	5	>40	80	10
cd4-CC	0.625	2.5	1.25	1.25	2.5	2.5	1.25	5	5	1.25
cd4-CCPP	0.625	1.25	1.25	1.25	2.5	1.25	0.625	2.5	5	0.625
** *S. aureus* **										
LL-37	1.25	>10	2.5	1.25	2.5	5	2.5	>40	>80	10
KR-12	10	>10	10	>10	>10	>10	10	>40	>80	>20
KR-12 (Q5K,D9A)	1.25	>10	2.5	>10	>10	5	2.5	>40	>80	10
cd4-CC	0.625	2.5	2.5	0.625	1.25	1.25	1.25	20	>40	1.25
cd4-CCPP	0.625	2.5	1.25	0.625	0.625	0. 625	0.625	10	>40	0.625
** *C. albicans* **										
LL-37	2.5	>10	2,5	2,5	>10	>10	2.5	np ^4^	np	np
KR-12	10	>10	10	>10	>10	>10	10	np	np	np
KR-12 (Q5K,D9A)	1.25	>10	10	10	>10	>10	5	np	np	np
cd4-CC	0.625	5	1.25	1.25	2.5	2.5	1.25	np	np	np
cd4-CCPP	0.625	5	0.625	0.625	2.5	1.25	0.625	np	np	np

**^1^** The control minimum inhibitory concentration (MIC) was determined in a two-step microdilution assay in Tris buffer 10 mM without salts. The final concentrations of NaCl, NH_4_Cl, KCl, CaCl_2_, MgCl_2_, and FeCl_3_ were 150 mM, 6 μM, 4.5 μM, 2.5 mM, 1 mM, and 4 μM, respectively. ^2^ MHB, Mueller–Hinton broth (rich media, unbuffered). ^3^ IE, ionic environment (50 mM NaHCO_3_, buffered MHB). ^4^ np, not performed.

**Table 2 biomedicines-11-00504-t002:** Minimum inhibitory concentrations (MICs) for AMP-resistant strains in buffer conditions.

Peptides	Bacterial Species and Strains					
	*S. typhimurium*			*E. coli*		
	WT ^1^	*waaY*	phoP	WT ^1^	DA54114	DA57107
LL-37	2.5	5	10	1	1	2.25
KR-12	20	20	20	1	1	1
KR-12 (Q5K,D9A)	10	10	10	1	1	1
cd4-CCPP	2.5	1.25	1.25	0.37	0.25	0.25
cd4-CC	1.25	2.5	1.25	-	-	-

^1^ WT, wild type.

## Data Availability

The data presented in this study are available on request from the corresponding author.

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
