# Peer review of "Transforming Cross-Linked Cyclic Dimers of KR-12 into Stable and Potent Antimicrobial Drug Leads"

_biomedicines, 2023, doi:10.3390/biomedicines11020504_

Round 1

Reviewer 1 Report

Design and development of analogs of AMPs could be useful to fight against MDR bacterial strains. In this manuscript, authors have designed novel cyclic analogs of LL37 driven peptide fragment KR12. KR12 appears to be weakly active in killing bacterial strains. The cyclic dimeric peptide demonstrated much higher activity in various media, including salt, and was found to be stable in serum compared to the parent peptide. This manuscript may be suitable for publication in the journal after a revision as outlined below.

1.    What was the rationale of using Tris buffer for MIC assays? Mostly, MIC assays are done using MH broth.

2.    NMR data of the peptide appears to be difficult for analysis, even for the free peptide. Structures of LL37 and KR12 could be determined in various solution conditions. Does the peptide aggregate in solution? The mode of action of the peptide could not be clearly established. Does the higher antibacterial activity of the peptide arise from high cationicity and better membrane interactions?

3.    For Gram-negative bacteria, LPS-AMPs interactions are vital in cell killing as demonstrated in helical peptides (J Am Chem Soc. 2010 132(51):18417-28. doi: 10.1021/ja1083255, Chemistry. 2009;15(9):2036-40. doi: 10.1002/chem.200802635, Biochemistry. 2015 17;54(10):1897-907. doi: 10.1021). Authors should address interactions of the peptide with LPS/outer membrane.

4.    Observation of antibacterial activity of the peptide in presence of salt is interesting. Although, why does the peptide is able to display such salt resistance activity is not clear. Helical salt resistant AMPs are reported in literature and mode of action could be elucidated (J Bacteriol. 2022 204(12):e0031222. doi: 10.1128/jb.00312-22, Biopolymers. 2016 106(3):345-56. doi: 10.1002/bip.22819). Authors can include these works in Discussion to better understand implication of the results.

5.    Are these designed peptide toxic to human cells?

Author Response

We thank the Editor and Reviewers for the overall positive response, and for the comments and suggestions that have helped to improve the manuscript.

Please find below a list of answers and actions based on the Reviewer’s comments.

Response to Reviewer 1 Comments

  1. What was the rationale of using Tris buffer for MIC assays? Mostly, MIC assays are done using MH broth. 

Response: The choice of assays is discussed in the manuscript in the results section (page 8, lines 204-215), and discussion section (page 11) and we have referred to our previous work where this is explained in more detail (ref 18 MS). It should also be noted that peptides in focus are tested active in MH broth.

  1. NMR data of the peptide appears to be difficult for analysis, even for the free peptide. Structures of LL37 and KR12 could be determined in various solution conditions. Does the peptide aggregate in solution? The mode of action of the peptide could not be clearly established. Does the higherantibacterial activity of the peptide arise from high cationicity and better membrane interactions? 

Response: It was not possible to assign the chemical shifts due to overlapping signals in the solution. Even the addition of lipids (SDS or lyso-PG) did not improve the signal dispersion. Even though parts of the structure is helical (as evident by CD), the disordered regions, likely coming from the connecting linker regions, make it difficult to assign individual chemical shifts when NMR is used. LL-37 is a very well-studied peptide and the NMR structure has been reported both in solution and micelles. We have reported KR-12 and derivatives structure in SDS-micelles (ref 16 MS), KR-12 adopts well-defined alpha-helical conformations in membrane-simulated SDS environments while in aqueous environment showed limited dispersion of peaks, indicative of unstructured conformations.

These peptides are readily soluble in water, so aggregation was not an issue. This has now been added to the results section between lines 91-93.

The higher antibacterial activity of cross-linked cyclic dimers is not just due to the overall positive charge but the high local concentration resulting from dimerization has also a role.

  1. For Gram-negative bacteria, LPS-AMPs interactions are vital in cell killing as demonstrated in helical peptides (J Am Chem Soc. 2010 132(51):18417-28. doi: 10.1021/ja1083255, Chemistry. 2009;15(9):2036-40. doi: 10.1002/chem.200802635, Biochemistry. 2015 17;54(10):1897-907. doi: 10.1021). Authors shouldaddressinteractions of the peptide with LPS/outer membrane.

Response: We have now added a paragraph in the discussion on pages 9-10 lines 313 to 322 to address the possible mechanism of action of dimers.

  1. Observation of antibacterial activity of the peptide in presence of salt is interesting. Although, why does the peptide is able to display such salt resistance activity is not clear. Helical salt resistant AMPs are reported in literature and mode of action could be elucidated (J Bacteriol. 2022 204(12):e0031222. doi: 10.1128/jb.00312-22, Biopolymers. 2016 106(3):345-56. doi: 10.1002/bip.22819). Authors can include these works in Discussion to better understand implication of the results. 

Response: In our hands, both the monomer peptides lost antimicrobial activities in the presence of different types of salts but the activities were marginally affected for LL-37 and cross-linked cyclic dimers against both Gram-negative and Gram-positive bacterial species. In monomer peptides including LL-37, N-and C-terminals are free and have more flexible skeletal which caused them to be more susceptible to salt ions compared to the cross-linked cyclic dimers.

  1. Are these designed peptide toxic to human cells?

Response: We have tested the hemolytic activity of these peptides against human red blood cells (RBCs) in the current manuscript and the relevant information is in the result section between lines 242 to 245. We have already reported the cytotoxicity of the dimers from the previous generations on normal human cells, RBCs, and lymphoma cell lines in our published papers and we expect the peptides in the current manuscript to fall in the same window (ref 16 and 17 in MS).

Author Response

We thank the Editor and Reviewers for the overall positive response, and for the comments and suggestions that have helped to improve the manuscript.

Please find below a list of answers and actions based on the Reviewer’s comments.

  • In Introduction section (lines 32-32, page 1) description of human cathelicidin should be more precise (for instance please see: Sørensen, O.E.; Gram, L.; Johnsen, A.H.; Andersson, E.; Bangsbøll, S.; Tjabringa, G.S.; Hiemstra, P.S.; Malm, J.; Egesten, A.; Borregaard, N. Processing of seminal plasma hCAP-18 to ALL-38 by gastricsin: a novel mechanism of generating antimicrobial peptides in vagina. J. Biol. Chem. 2003, 278(31), 28540-28546. doi:10.1074/jbc.M301608200.)

Response: We acknowledge the reviewer's suggestion, and now we have included more precise information about LL-37 (introduction lines 33-34) and also included relevant references.

  • Page 2, figure 1 should include apart from 3D structures of investigated peptides also their sequences. In cyclic form disulfide bond and place of head-to-tail cyclization should be indicated.

Response: We highly appreciate the reviewer's comment. Figure 1 has been modified to include the amino acid sequences of the peptides synthesized and the place of head-to-cyclization has also been highlighted (Introduction, Figure 1  lines 75-76).

  • P_a_g_e_ _4_,_ _l_i_n_e_ _1_2_9_,_ _i_t_ _s_h_o_u_l_d_ _b_e_ _4_B_ _i_n_s_t_e_a_d_ _o_f_ _“4_A_”._ _

Response: The text has been corrected to state that 4A should be 4B.

  • Page 5, Table 1, It should be KR-12 instead of “K2-12”.

Response: The text has been corrected to state that K2-12 should be KR-12 in Table 1.
